# Liensinine Prevents Acute Myocardial Ischemic Injury via Inhibiting the Inflammation Response Mediated by the Wnt/β-Catenin Signaling Pathway

**DOI:** 10.3390/ijms26104566

**Published:** 2025-05-10

**Authors:** En Ma, Jingwei Zhang, Yirong Tang, Xue Fang, Canran Wang, Celiang Wu, Weidong Zhu, Da Wo, Dan-ni Ren

**Affiliations:** Academy of Integrative Medicine, College of Integrative Medicine, Fujian University of Traditional Chinese Medicine, Fujian Key Laboratory of Integrative Medicine on Geriatric, 1 Qiuyang Road, Minhou, Fuzhou 350122, China; 18217497624@163.com (E.M.); 15803856395@163.com (J.Z.); trong750@163.com (Y.T.); 15705961231@163.com (X.F.); 18437850478@163.com (C.W.); 15821726396@163.com (C.W.); wzhu@tongji.edu.cn (W.Z.)

**Keywords:** myocardial infarction, liensinine, inflammation, Wnt/β-catenin

## Abstract

Myocardial infarction (MI) is characterized by the sudden reduction in myocardial blood flow and remains the leading cause of death worldwide. Because MI causes irreversible damage to the heart, discovering drugs that can limit the extent of ischemic damage is crucial. Liensinine (LSN) is a natural alkaloid that has exhibited beneficial effects in various cardiovascular diseases, including MI; however, its molecular mechanisms of action remain largely unelucidated. In this study, we constructed murine models of MI to examine the potential beneficial effects and mechanisms of LSN in myocardial ischemic injury. Murine models of MI in wild-type and cardiomyocyte-specific β-catenin knockout mice were used to explore the role of LSN and Wnt/β-catenin signaling in MI-induced cardiac injuries and inflammatory responses. The administration of LSN markedly improved cardiac function and decreased the extent of ischemic damage and infarct size following MI. LSN not only prevented excessive inflammatory responses but also inhibited the aberrant activation of Wnt/β-catenin signaling, two factors that are critically involved in the exacerbation of MI-induced injury. Our findings provide important new mechanistic insight into the beneficial effect of LSN in MI-induced cardiac injury and suggest the therapeutic potential of LSN as a novel drug in the treatment of MI.

## 1. Introduction

Myocardial infarction (MI) is a result of coronary artery thrombosis or occlusion and remains the leading cause of death worldwide [1,2,3,4]. Upon the incidence of MI, the injured myocardium triggers an inflammatory response, which plays a vital role in the ischemic heart [5]. A major hallmark of MI is activation of an inflammatory response, resulting in a massive infiltration of immune cells and the release of cytokines that play a role in wound healing [6,7]. However, prolonged and exaggerated inflammatory response following MI leads to adverse cardiac remodeling and eventually heart failure [6,8]. Although various drugs such as adrenergic receptor blockers, calcium channel blockers, and renin–angiotensin system inhibitors are clinically used to assist in the treatment of myocardial infarction, no current drugs can reverse cardiac damage once MI has already occurred, since damage to the myocardium in MI is irreversible [9,10]. Thus, the discovery of novel drug candidates for the treatment of pathological remodeling and inflammatory response following myocardial infarction is critical.

The Wnt/β-catenin signaling pathway plays a pivotal role in development, cardiomyocyte differentiation, apoptosis, and disease progression [11,12,13,14]. There is increasing evidence suggesting that Wnt signaling is aberrantly activated during the pathological process of myocardial infarction [4,15,16]. The inhibition and deletion of the Wnt/β-catenin signaling pathway have been shown to mitigate pathological remodeling during myocardial infarction [17,18]. Further, there appears to be a complex and vital crosstalk mechanism between Wnt/β-catenin signaling and inflammatory pathways involved in the pathogenesis of MI [19]. Therefore, novel therapeutic applications that involve inhibiting MI-induced inflammatory response and Wnt/β-catenin signaling may be effective for alleviating cardiac ischemic injury.

Liensinine (LSN) is a natural bisbenzylisoquinoline alkaloid that is derived from mature lotus (Nelubonaceae) seeds [20,21]. Studies have shown that LSN exhibits multiple beneficial effects, including anti-oxidative, anti-inflammatory, and anti-hypertensive effects in cardiovascular diseases [20,22]. However, the potential mechanisms of LSN in preventing ischemic cardiac injuries remain largely unelucidated. In our present study, we aimed to examine the protective effects of LSN in a mouse model of myocardial infarction and elucidate the potential mechanism of LSN in preventing myocardial ischemic injury and adverse inflammatory response via inhibition of the Wnt/β-catenin pathway.

## 2. Results

### 2.1. LSN Protects Against MI-Induced Cardiac Ischemic Injury in Mice

To study the potential beneficial effects of LSN in the ischemic myocardium, we first constructed mouse models of MI via the ligation of the left anterior descending coronary artery and subsequently administered equal volumes of either PBS or LSN. Cardiac function was assessed by echocardiography four weeks after MI, which showed that MI induced significant decreases in left ventricular (LV) ejection fraction (EF%) and fractional shortening (FS%) parameters, which were markedly improved in mice administered with LSN (Figure 1A). Moreover, M-mode echocardiographic assessment showed that the MI-induced deterioration of LV wall movement was improved in LSN-administered mice (Figure 1B). The list of cardiac function parameters was summarized, demonstrating that LSN administration resulted in an overall improvement in heart function following MI (Figure 1C).

To further examine the cardiac protective effect of LSN following myocardial infarction, we examined the differences in infarct size 4 weeks after MI. Masson’s trichrome staining showed that LSN-administered mice had significantly reduced infarct size and collagen fiber deposition compared to PBS-administered mice (Figure 1D). In addition, serum levels of hypertrophic biomarkers, including ANP, BNP, and cardiac troponin I (CTnI), were markedly increased post-MI, which were also significantly attenuated in mice administered with LSN (Figure 1E). Real-time PCR analysis showed that the mRNA levels of fibrotic markers Collagen-I (Col-I) and Collagen-III (Col-III), as well as cardiac hypertrophic markers ANP, BNP, and β-MHC, also exhibited a similar trend following MI (Figure 1F). These results collectively demonstrate the robust protective effect of LSN in cardiac ischemic injury.

### 2.2. LSN Prevents MI-Induced Early Cardiac Inflammatory Response

It is well established that exaggerated inflammatory response promotes adverse cardiac remodeling and tissue injury [6]. To determine whether the beneficial effect of LSN was due to preventing adverse inflammatory response following MI, we examined the levels of inflammatory biomarkers at 7 days post-MI both in the infarcted myocardium and serum. Real-time PCR showed that the levels of proinflammatory cytokines TNF-α, IL-6, and IL-1β were dramatically upregulated in the infarcted myocardium 7 days post-MI, which were significantly reduced in mice administered with LSN (Figure 2A). Serum levels of TNF-α, IL-6, IL-1β, and IFN-γ also exhibited a similar pattern (Figure 2B), demonstrating that LSN administration can robustly attenuate MI-induced proinflammatory responses (Figure 2B).

Of note, NF-κB is well known as a master regulator of the inflammatory response, whose activation and translocation to the nucleus leads to the expression of interleukins and cytokines, which mediate MI-induced maladaptive LV remodeling and functional deterioration. Hence, we further examined the expression of NF-κB in the infarcted myocardium via immunoblot and immunofluorescence staining. MI induced the apparent phosphorylation of the P65 subunit of NF-κB and IκB in the infarcted myocardium at 1 week post-MI, which was significantly attenuated in mice administered with LSN (Figure 2C). Moreover, MI induced apparent NF-κB nuclear translocation, which was also prevented in LSN-administered mice (Figure 2D). Taken together, these results demonstrate that LSN can strongly prevent MI-induced NF-κB signaling activation and early cardiac inflammatory responses.

### 2.3. β-Catenin Deletion Prevents MI-Induced Cardiac Injury and Inflammatory Response

The crosstalk between Wnt/β-catenin and NF-κB signaling has been shown to play a vital role in MI-induced inflammation [19]. Hence, we further examined the effects of cardiac-specific deletions of β-catenin in the heart via immunoblotting analysis (Appendix A). At one week post-MI, β-catenin KO mice exhibited significant improvements in ejection fraction and fractional shortening compared to β-cateninfl/fl floxed littermates following MI (Figure 3A). The summary of cardiac function parameters showed an overall improvement of heart function in β-catenin KO following MI (Figure 3B). M-mode assessment further showed a marked improvement in LV wall movement in β-catenin KO mice compared to floxed littermates (Figure 3C). Moreover, the elevation of serum biomarkers ANP, BNP, and CTnI was markedly attenuated in β-catenin KO mice at 1 week post-MI (Figure 3D). These results indicate that inhibition of Wnt/β-catenin is protective in MI-induced cardiac injury.

In addition, MI-induced phosphorylation of NF-κB P65 and iκB in the infarcted myocardium at 1 week post-MI was also significantly prevented in β-catenin KO mice compared to floxed littermates (Figure 3E). Similarly, the nuclear translocation of NF-κB following MI was also prevented in β-catenin KO mice (Figure 3F). We further examined whether the effects of LSN in MI-induced inflammatory response had any relationship to its ability to inhibit Wnt/β-catenin signaling. Notably, MI induced the upregulation of nuclear P65 and β-catenin but the downregulation of cytoplasmic phospho-β-catenin in the infarcted myocardium at 1 week post-MI, demonstrating the activation of Wnt/β-catenin and NF-κB signaling post-MI (Figure 3G). Mice administered with LSN significantly attenuated these expressions, indicating that LSN inhibits Wnt/β-catenin signaling by promoting β-catenin phosphorylation and subsequent degradation (Figure 3G). These results demonstrate that the inhibition of Wnt/β-catenin signaling is beneficial in preventing MI-induced NF-κB signaling activation and inflammatory response.

### 2.4. LSN Prevents Inflammatory Response via Inhibiting Wnt/β-Catenin Signaling

Tumor necrosis factor α (TNF-α) is a key activator of the inflammatory response that is highly upregulated in various diseases, including MI. We thus used TNF-α to mimic the occurrence of an inflammatory response in vitro. Indeed, treatment with TNF-α for 12 h resulted in obvious increases in the levels of P-P65, P-iκB, nuclear β-catenin, and P65 but a significant decrease in cytoplasmic phospho-β-catenin in rat H9C2 and human AC16 cardiomyocytes, which were prevented in cells pretreated with LSN (Appendix A). These findings indicate that LSN prevents TNF-α-induced Wnt/β-catenin activation by promoting β-catenin phosphorylation and subsequent degradation. Immunofluorescence staining further demonstrated distinct nuclear localization of P65 following TNF-α treatment, which was largely prevented in cells pretreated with LSN (Figure 4A). Moreover, siRNA knockdown of β-catenin significantly decreased TNF-α-induced upregulations of P-P65, P-iκB, nuclear β-catenin, and P65 (Figure 4B,C and Appendix A). These results suggest that the inhibition of Wnt/β-catenin signaling prevents TNF-α-induced inflammatory response.

To further investigate the role of Wnt/β-catenin signaling in MI-induced inflammatory response, we used LiCl, a stabilizer of β-catenin that prevents intrinsic cytoplasmic β-catenin degradation. Indeed, LiCl treatment significantly induced β-catenin accumulation and the upregulation of P-P65, P-iκB, and nuclear P65 accumulation in rat H9C2 and human AC16 cardiomyocytes (Figure 4D,E and Appendix A). Of note, co-treatment with TNF-α further enhanced the levels of these proteins (Figure 4D,E and Appendix A). Taken together, these results demonstrate that the beneficial effects of LSN in preventing cardiac ischemic and inflammation-induced cardiomyocyte injury were via the inhibition of the Wnt/β-catenin signaling pathway.

## 3. Discussion

One of the key features of myocardial infarction is the infiltration of immune cells and the subsequent inflammation response [23,24]. Numerous studies have demonstrated that NF-κB signaling is one of the most important pathways responsible for the regulation of MI-induced inflammation [25,26,27,28]. Thus, the timely resolution of adverse NF-κB activation would be beneficial following myocardial ischemia. Hence, our present study, demonstrating the robust effect of LSN in inhibiting MI-induced NF-κB activation and inflammatory response, provides important insight that supports the therapeutic potential of LSN in the treatment of MI.

Increasing evidence has shown that Wnt/β-catenin signaling activation is associated with worsened progression of MI, while, in contrast, the inhibition of Wnt signaling actually mitigates myocardial injuries and pathological remodeling [10,17,29,30,31]. Importantly, Wnt/β-catenin signaling inhibition has been shown to be critical for the modulation of MI-induced inflammatory responses [19,32,33]. Interestingly, membrane lipid rafts play crucial roles in the modulation of the Wnt/β-catenin pathway, and therapeutics that target lipid rafts have become a growing strategy in the treatment of various diseases, including cardiovascular disease. In particular, enhanced endocytosis of the lipid raft-localized Wnt-receptor LRP6 suppresses Wnt/β-catenin signaling, while cells pretreated with lipid raft inhibitor methyl-β-cyclodextrin disrupt lipid rafts and prevent Wnt signal transduction [34,35,36,37]. Our previous study showed that LSN exerts protective effects during MI through the inhibition of Wnt/β-catenin signaling activation [29]. Whether LSN also exerts its effects due to modulation of lipid rafts warrants further investigation.

In our present study, we further demonstrated that the pharmacological effects and mechanisms of LSN in protecting against MI were via preventing inflammation-induced cardiomyocyte injury and inhibiting adverse Wnt/β-catenin signaling activation. In addition, the inhibition of Wnt/β-catenin signaling prevented MI-induced pathological cardiac remodeling and inflammatory responses. Due to complex crosstalk between the NF-κB and Wnt/β-catenin signaling pathways, drugs that can target one or both pathways may be an important and novel option in treating MI-induced inflammatory and ischemic injury. In this regard, our present study, showing the strong ability of LSN in not only inhibiting the adverse activation of Wnt/β-catenin signaling but also preventing the exacerbation of MI-induced inflammatory response, suggests a potential novel strategy for the therapeutic treatment of acute myocardial infarction. This multi-target effect of LSN may also address limitations in the current standards of care and treatments in inhibiting excessive inflammatory responses following MI. Future investigations that focus on elucidating LSN’s clinical dosing regimens, safety profile, and synergistic effects with existing therapeutic approaches are required in order to facilitate its translation into clinical applications.

Taken together, these findings elucidate the robust pharmacological effect and mechanisms of LSN in preventing myocardial ischemic injury and provide a theoretical basis for the potential use of LSN as a novel drug in the therapeutic treatment of MI.

## 4. Materials and Methods

### 4.1. Animal Model

All animals were housed in a temperature-controlled animal facility with a 12 h light/dark cycle and had free access to tap water and rodent chow. The study protocol was carried out under the code of practice for the ethical care and use of laboratory animals by the Fujian University of Traditional Chinese Medicine Laboratory Animal Care and Use Committee, Fujian, China. (No. FJTCM IACUC 2024246; approval date: 19 January 2025). Ten-week-old wild-type C57BL/6J male mice were used for models of myocardial infarction and MI as previously described [29]. All animal surgical procedures and subsequent analysis were performed by a blinded investigator. For experiments involving administration with LSN (MedChemExpress, Monmouth Junction, NJ, USA), mice were randomly divided into 3 groups: (1) sham operation group (i.p.); (2) MI + PBS group (i.p.); (3) MI + LSN (10 mg/kg/day, i.p.); *n* = 11 or more for each group.

### 4.2. Two-Dimensional Echocardiography

Two-dimensional echocardiography was performed using the Visual Sonics Vevo 2100 Imaging System by a blinded investigator according to standard protocol. Mice were anesthetized via inhaled isoflurane (Sigma Aldrich, St. Louis, MO, USA) using a vaporizer (EZ Anesthesia, Cumming, GA, USA). Depth of anesthesia was controlled as 1% isoflurane, which resulted in a heart rate of approximately 450 beats per minute for all subsequent measurements. M-mode measurement was used to determine left ventricular (LV) dimensions, including left ventricular internal dimension at the end diastole (LVID;d) and systole (LVID;s). LV ejection fraction (EF%) was calculated based on the following formula: (LVIDd3 − LVIDs3)/LVIDd3 × 100 (%).

### 4.3. Quantitative Real-Time Polymerase Chain Reaction (qRT-PCR)

Total RNA in cells and heart tissues was extracted using TRIzol reagent (Takara Biotechnology, Tokyo, Japan), and the cDNA was synthesized using the cDNA Synthesis kit (Takara Biotechnology, Tokyo, Japan). qRT-PCR was performed with SYBR Green Master Mix (Applied Biosystems, Carlsbad, CA, USA) to examine the relative mRNA expression levels of the indicated genes.

### 4.4. Cell Culture and siRNA Knockdown

Adult human AC16 cardiomyocytes and rat H9C2 cardiomyocytes were used for in vitro experiments. Cells were seeded on 35 mm dishes using high glucose DMEM (Gibco Thermo Fisher Scientific, San Diego, CA, USA) and cultured in a 37 °C incubator containing 5% CO_2_. For co-treatment with LSN and TNF-α, cells were pretreated with 0.5 μM LSN for 24 h prior to treatment with 10 ng/mL TNF-α for 12 h to model myocardial inflammatory response. For LiCl-induced β-catenin activation, cells were pretreated with LiCl for 4 h prior to treatment with 10 ng/mL TNF-α for 12 h. For the knockdown of β-catenin, cells were first transfected with RNAiMAX and β-catenin siRNA oligonucleotides or negative control siRNA (Thermo Scientific, Waltham, MA, USA) as a control in OPTI-MEM (Gibco Thermo Fisher Scientific) for 48 h.

### 4.5. Histology

Histological analyses were performed according to standard protocol. Briefly, hearts were fixed with 4% paraformaldehyde (0.2 μm filtered, PH. 8.0) overnight, dehydrated, paraffin-embedded, and sectioned (5 μm). The degree of collagen deposition and myocyte cross-sectional areas were detected by Masson’s trichrome staining and photographed using light microscopy, and the resulting images were analyzed using a quantitative digital analysis system (Image-Pro Plus 6.0).

### 4.6. Immunofluorescence

Immunofluorescence staining was performed according to standard protocol. Briefly, samples were fixed with 4% ice-cold paraformaldehyde (PFA) (Sigma Aldrich, St. Louis, MO, USA), permeabilized with 0.25% Triton X-100 (Sigma, St. Louis, MO, USA), and then blocked in 5% BSA (Roche, Basel, Switzerland). Samples were then incubated overnight at 4 °C with the following primary antibodies: Actn2 (1:800, Sigma, St. Louis, MO, USA), CTnT (1:800, Abcam, Cambridge, UK), or P65 (1:400, Cell Signaling Technology, Danvers, MA, USA). Next, samples were incubated with the respective secondary antibodies conjugated to Alexa Fluor 488 and Alexa Fluor 647 (1:400, Abcam, Cambridge, UK) and mounted in DAPI fluorescence mounting medium (Beyotime, Shanghai, China).

### 4.7. Western Blot Analysis

Western blot analyses were performed according to standard protocol. Briefly, total and nuclear proteins were extracted by the total protein extraction kit (P0013F, Beyotime, Shanghai, China) and the nucleoprotein extraction kit (#C500009, Sangon Biotech, Shanghai, China), respectively. Proteins were separated on 10% SDS-PAGE gels and then transferred onto a 0.22 µm PVDF membrane, blocked with 5% non-fat milk, and incubated with the respective primary antibodies: anti-β-catenin (#8840, Cell Signaling Technology), anti-TBP (#8515, Cell Signaling Technology), anti-GAPDH (#60004-1, Proteintech, Chicago, IL, USA), anti-NF-κB P65 (#8242, Cell Signaling Technology), anti-phospho-P65 (#3033, Cell Signaling Technology), anti-phospho-iκB (#2859, Cell Signaling Technology), and anti-phosphor-β-catenin (#9561, Cell Signaling Technology) overnight at 4 °C. Subsequently, membranes were incubated with the HRP-conjugated secondary antibodies and the resulting protein bands were detected via chemiluminescence.

### 4.8. Statistical Analysis

Statistical analyses were performed using SPSS 26.0 software (Chicago, IL, USA). Data were expressed as the mean ± standard error of the mean (sem). Student’s *t*-test was used to compare differences between two groups, and one-way ANOVA analysis was used to compare differences between three or more groups, followed by Fisher’s LSD post hoc analysis. *p*-values < 0.05 were considered statistically significant.

## Figures and Tables

**Figure 1 ijms-26-04566-f001:**
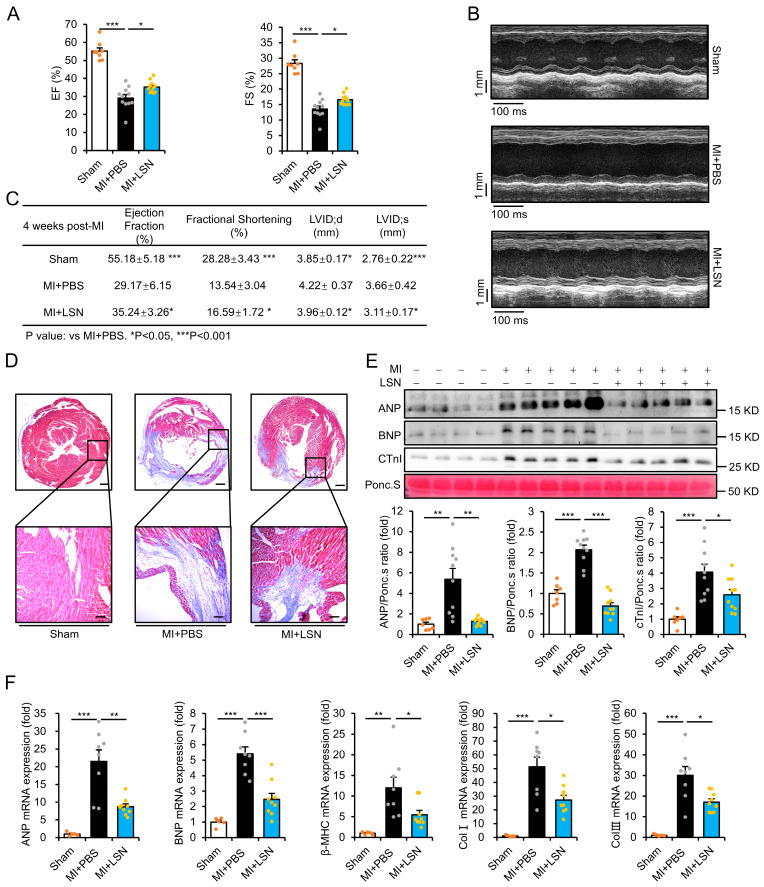
LSN prevents MI-induced cardiac dysfunction and remodeling in mice. (**A**) Echocardiographic assessment of ejection fraction (left) and fractional shortening (right) at 4 weeks post-myocardial infarction in mice administered with PBS or LSN. *n* = 8 or more. (**B**) M-mode echocardiography at 4 weeks post-myocardial infarction in mice administered with PBS or LSN. (**C**) Table of cardiac function parameters in Sham, MI + PBS, and MI + LSN at 4 weeks post-MI. Data are presented as mean ± SD. *n* = 8 or more for each group. LVID;d: Left Ventricular Internal Dimension, diastolic; LVID;s: Left Ventricular Internal Dimension, systolic. (**D**) Masson’s trichrome staining showing ischemic hearts (top, scale bar = 1000 µm) and enlarged area of local left ventricular (LV) border region of ischemic hearts 4 weeks post-MI (bottom, scale bar = 100 μm). (**E**) Representative immunoblots (upper) and quantification (lower) of serum ANP, BNP, and CTnI in mice administered with PBS or LSN at 3 weeks post-MI. *n* = 7 or more. (**F**) ANP, BNP, β-MHC, Collagen I, and Collagen III mRNA levels in heart tissue of mice administered with PBS or LSN at 1 week post-MI. *n* = 5 or more. Data are presented as mean ± SEM along with individual data points; * *p* < 0.05, ** *p* < 0.01, *** *p* < 0.001 vs. MI + PBS.

**Figure 2 ijms-26-04566-f002:**
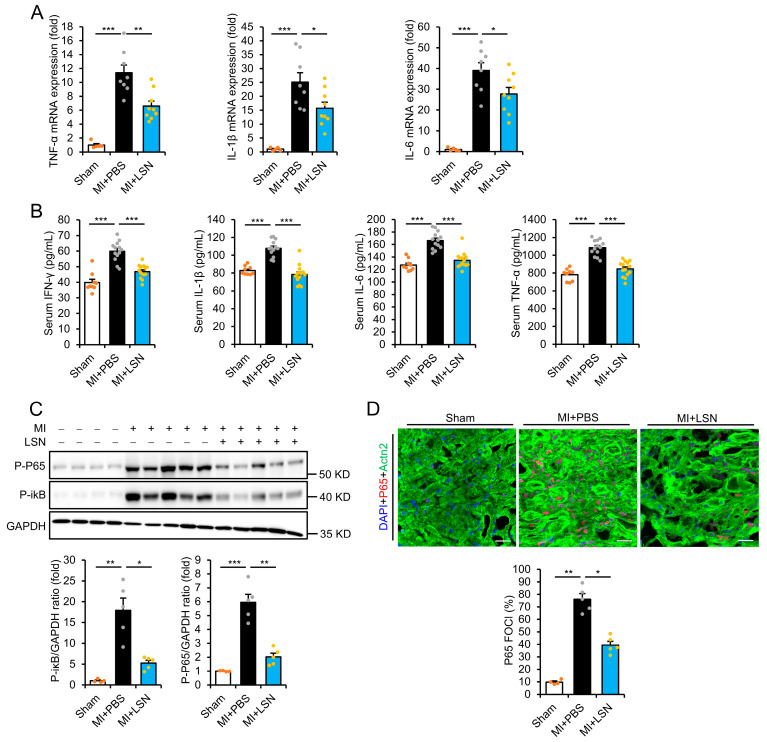
LSN prevents MI-induced early cardiac inflammatory responses. (**A**) TNF-α, IL-1β, and IL-6 mRNA levels in heart tissue of mice administered with PBS or LSN at 1 week post-MI. *n* = 5 or more. (**B**) ELISA analysis of serum IFN-γ, IL-1β, IL-6, and TNF-α levels in mice administered with PBS or LSN at 1 week post-MI. *n* = 9 or more. (**C**) Representative immunoblots (upper) and quantification (lower) of P-P65 and P-iκB expression in the heart of mice administered with PBS or LSN at 1 week post-MI. *n* = 4 or more. (**D**) Immunofluorescence staining (upper) and quantification (lower) of P65 foci (red), cardiomyocyte-specific α-actinin 2 (Actn2, green), and DAPI (blue) in heart tissue of mice treated as in (**C**). *n* = 4 or more. Scale bar, 16 μm. Data are presented as mean ± SEM along with individual data points; * *p* < 0.05, ** *p* < 0.01, *** *p* < 0.001 vs. MI + PBS.

**Figure 3 ijms-26-04566-f003:**
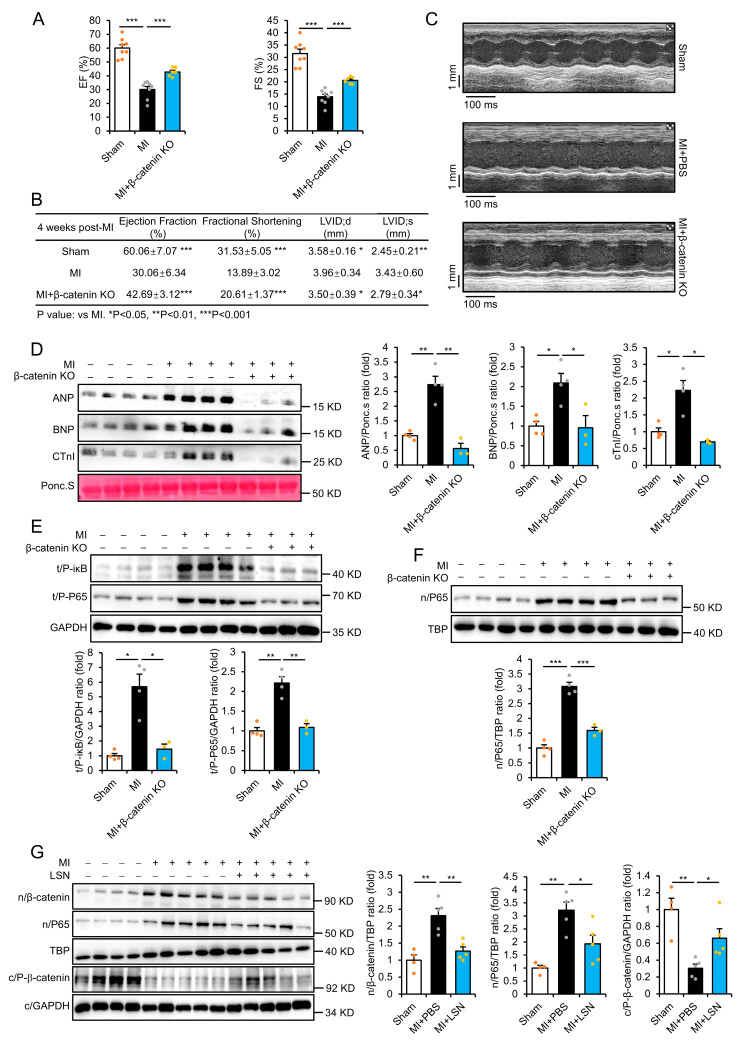
β-catenin deletion prevents MI-induced cardiac injury and inflammatory response. (**A**) Echocardiographic assessment of ejection fraction (left) and fractional shortening (right) at 1 week post-MI in β-catenin fl/fl and β-catenin KO mice. *n* = 6 or more. (**B**) Table of cardiac function parameters in β-catenin fl/fl and β-catenin KO mice at 1 week post-MI. *n* = 6 or more for each group. LVID;d: Left Ventricular Internal Dimension; diastolic; LVID;s: Left Ventricular Internal Dimension; systolic. (**C**) M-mode echocardiography at 1 week post-MI in β-catenin fl/fl and β-catenin KO mice. (**D**) Representative immunoblots (left) and quantification (right) of serum ANP, BNP, and cTnI in β-catenin fl/fl and β-catenin KO mice at 1 week post-MI. *n* = 3 or more. (**E**) Representative immunoblots (upper) and quantification (lower) of total P-P65 and P-iκB expression in the heart of β-catenin fl/fl and β-catenin KO mice at 1 week post-MI. *n* = 3 or more. (**F**) Representative immunoblots (upper) and quantification (lower) of nuclear P65 expression in the heart of β-catenin fl/fl and β-catenin KO mice at 1 week post-MI. *n* = 3 or more. (**G**) Representative immunoblots (left) and quantification (right) of n/nuclear β-catenin, P65, and c/cytoplasmic phospho-β-catenin expression in the heart of mice administered with PBS or LSN at 1 week post-MI. *n* = 4 or more. Data are presented as mean ± SEM along with individual data points; * *p* < 0.05, ** *p* < 0.01, *** *p* < 0.001.

**Figure 4 ijms-26-04566-f004:**
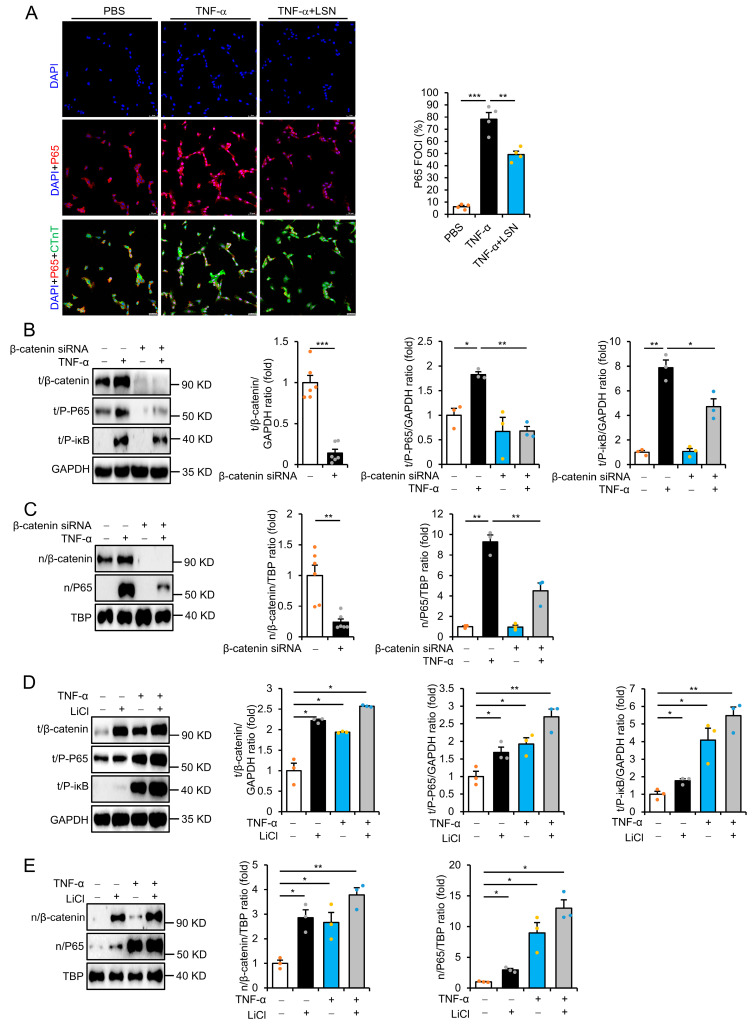
LSN prevents MI-induced proinflammatory response through inhibiting Wnt/β-catenin signaling. (**A**) Immunofluorescence staining (left) and quantification (right) of P65 foci (red), CTnT (green), and DAPI (blue) in H9C2 cardiomyocytes induced by TNF-α following pretreatment with LSN for 12 h. *n* = 4. Scale bar, 50 μm. (**B**) Representative immunoblots (left) and quantification (right) of total β-catenin, P-P65, and P-iκB expression following siRNA-mediated knockdown of β-catenin for 48 h, and subsequent treatment with TNF-α for 12 h in H9C2 cardiomyocytes. *n* = 3 or more. (**C**) Representative immunoblots (left) and quantification (right) of nuclear β-catenin and P65 expression following siRNA-mediated knockdown of β-catenin for 48 h, and subsequent treatment with TNF-α for 12 h in H9C2 cardiomyocytes. *n* = 3 or more. (**D**) Representative immunoblots (left) and quantification (right) of total β-catenin, P-P65, and P-iκB expression following TNF-α or LiCl treatment for 12 h in H9C2 cardiomyocytes. *n* = 3 or more. (**E**) Representative immunoblots (left) and quantification (right) of nuclear β-catenin and P65 expression following TNF-α or LiCl treatment for 12 h in H9C2 cardiomyocytes. *n* = 3 or more. Data are presented as mean ± SEM along with individual data points; * *p* < 0.05, ** *p* < 0.01, *** *p* < 0.001.

## Data Availability

The data from this study are available from the corresponding author upon reasonable request.

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
