# Peer review of "Liensinine Prevents Acute Myocardial Ischemic Injury via Inhibiting the Inflammation Response Mediated by the Wnt/β-Catenin Signaling Pathway"

_ijms, 2025, doi:10.3390/ijms26104566_

Round 1
Reviewer 1 Report
Comments and Suggestions for Authors
The manuscript by En Ma et al. investigates the cardioprotective effects of Liensinine (LSN), a naturally occurring bisbenzylisoquinoline alkaloid, in the context of myocardial infarction (MI). Utilizing both wild-type and cardiomyocyte-specific β-catenin knockout mouse models, the authors present convincing data demonstrating that LSN alleviates cardiac dysfunction and dampens the inflammatory response following MI, primarily via suppression of the Wnt/β-catenin/NF-κB signaling pathway. The study is thoughtfully designed and contributes meaningful insights to the expanding body of research on therapeutic strategies for ischemic heart disease.
If possible, the authors could assess the levels of phosphorylated β-catenin (e.g., at Ser33/37/Thr41), which are typically associated with β-catenin degradation and Wnt pathway inactivation. This analysis would strengthen the evidence that LSN inhibits Wnt/β-catenin signaling by promoting β-catenin phosphorylation and subsequent degradation
The Wnt pathway has been extensively studied in the context of membrane lipid rafts. I would appreciate it if the authors could take this into consideration by adding also a review of the early 2023 on the advances and molecular mechanisms and signaling through lipid rafts, as well as some recent experimental studies highlighting the involvement of Wnt signaling (LRP6 and β-catenin ) through lipid rafts and the reduction of signaling through the use of lipid raft inhibitors such as beta-methyl-cyclodextrin
Reviewer 2 Report
Comments and Suggestions for Authors
Congratulations on your well-executed and compelling manuscript. The research presented is methodologically robust and offers substantial mechanistic evidence for the cardioprotective effects of Liensinine (LSN) in the context of myocardial infarction. The link between LSN modulation of the Wnt/β-catenin and NF-κB signaling pathway is well described. I also found the integration of both in vivo and in vitro methodologies to be valuable to enhances the validity of the drawn conclusions. Nevertheless, I think in the future studies, author should consider preforming chromatin immunoprecipitation (ChIP)in order to assess whether NF-κB binding to the promoters of key proinflammatory genes. That would provide direct evidence of transcriptional regulation by this pathway. Also, author can investigate the modulation of upstream Wnt ligands (such as Wnt1 and Wnt3a) or receptors (including members of the Frizzled family) by LSN, which could elucidate its specific point of intervention within the Wnt/βatenin signaling pathway. This exploration critical for defining the precise through which LSN exerts its effects on this pathway.
These studies are for future consideration, for this manuscript I would like to recommend adding the translational relevance of your findings, as this would provide insights into the potential clinical implications and applications of LSN in therapeutic contexts. Addressing issues such as potential therapeutic application, dosing, safety, and comparability to current standards of care would significantly strengthen the impact and clinical perspective of your manuscript.
Round 2
Reviewer 1 Report
Comments and Suggestions for Authors
the authors have answered to my requests